# Fusion Expression and Immune Effect of PCV2 Cap Protein Tandem Multiantigen Epitopes with CD154/GM-CSF

**DOI:** 10.3390/vetsci8100211

**Published:** 2021-09-29

**Authors:** Qian Mao, Weijian Zhang, Shengming Ma, Zilong Qiu, Bingke Li, Chen Xu, Huangyu He, Shuangqi Fan, Keke Wu, Jinding Chen, Mingqiu Zhao

**Affiliations:** 1Department of Microbiology and Immunology, College of Veterinary Medicine, South China Agricultural University, Guangzhou 510642, China; maoqian@stu.scau.edu.cn (Q.M.); 20203073166@stu.scau.edu.cn (W.Z.); mashengming@stu.scau.edu.cn (S.M.); qiuzilong@stu.scau.edu.cn (Z.Q.); 20193073038@stu.scau.edu.cn (B.L.); 97xc@stu.scau.edu.cn (C.X.); hehuangyu@stu.scau.edu.cn (H.H.); shqfan@scau.edu.cn (S.F.); wukeke@stu.scau.edu.cn (K.W.); jdchen@scau.edu.cn (J.C.); 2Key Laboratory of Zoonosis Prevention and Control of Guangdong Province, Guangzhou 510642, China

**Keywords:** porcine circovirus type 2, capsid protein, subunit vaccine, protective immunity

## Abstract

Porcine circovirus associated diseases (PCVAD) is a contagious disease of swine caused by porcine circovirus type 2 (PCV2). The capsid protein (Cap) is the sole structural protein and the main antigen of PCV2. Cap is the principal immunogenic protein and induces humoral and cellular immunity. CD154 and GM-CSF are immune adjuvants that enhance responses to vaccines. However, whether these two cellular molecules could produce an enhanced effect in PCV2 vaccines still needs to be further studied. The results of PCR and restriction enzyme showed that the recombinant lentiviral plasmids pCDH-TB-Cap, pCDH-TB-Cap-CD154 and pCDH-TB-Cap were successfully constructed. Western blot and IFA showed that the three fusion proteins TB-Cap, TB-Cap-CD154 and TB-Cap-GM-CSF were stably expressed in CHO-K1 cells. Indirect ELISA assay showed that mice immunized with TB-Cap-CD154 and TB-Cap-GM-CSF fusion proteins produced higher PCV2-specific antibodies than mice immunized with the TB-Cap and a commercial vaccine (*p* < 0.0001). Moreover, lymphocyte proliferation and flow cytometry showed that the cellular immune response of each immune group was significantly enhanced (*p* < 0.0001). After PCV2 challenge, the results revealed that the viral loads in serum, lung and kidney of all vaccinated groups were significantly lower than the PBS group (*p* < 0.0001). The transcription levels of IL-2, IFN-gamma, IL-4 and IL-10 cytokines in the TB-Cap, TB-Cap-CD154 and TB-Cap-GM-CSF groups were significantly higher than those in the PBS and recombinant vaccine groups (*p* < 0.0001). These results indicated that CD154 and GM-CSF could enhance the ability of TB-Cap protein to induce the body to produce PCV2 specific antibodies and increase the transcription level of cytokines. Thus, CD154 and GM-CSF molecules were a powerful immunoadjuvant for PCV2 subunit vaccines. The novel TB-Cap-CD154 and TB-Cap-GM-CSF subunit vaccine has the potential to be used for the prevention and control of PCVAD.

## 1. Introduction

Porcine circovirus (PCV) is the smallest known animal virus, belonging to the genus *Circovirus* in the *Circoviridae* family and can be divided into three serotypes, Porcine circovirus type 1 (PCV1), Porcine circovirus type 2 (PCV2) and Porcine circovirus type 3 (PCV3) [1]. PCV1 is a non-pathogenic virus, PCV2 and PCV3 are pathogenic viruses [2]. Recently, a novel circovirus was identified and tentatively designated as porcine circovirus type 4 (PCV4) [3]. PCV2 infection in pigs causes a series of Porcine circovirus associated diseases (PCVAD), such as postweaning multisystemic wasting syndrome (PMWS), porcine dermatitis nephropathy syndrome (PDNS) and swine reproductive disorder syndrome (SRDS) and other varieties of clinical syndromes, which brings great economic losses to the global pig industry [4,5]. At present, there is no specific drug for PCVAD. Therefore, effective preventing measures will play a vital role in controlling the spread of the PCV2. In recent years, the main types of PCV2 commercial vaccines on the market are inactivated vaccine and subunit vaccine [6]. Inactivated vaccines are antigenic but lacked the capability to stimulate the body to produce complete cellular immune responses. The current PCV2 subunit vaccines are often poorly immunogenic as they represent a small portion of the pathogen and may require multiple vaccinations with large doses to provide protective immunity [7]. In order to effectively prevent and control PCVAD, there is an urgent need to develop a new safe, efficient and cheap vaccine. Among them, the development of enhancing the immunogenicity of the PCV2 subunit vaccine is particularly important. PCV2 capsid (Cap) protein is a viral nucleocapsid protein encoded by open reading frame 2 (ORF2) and contains specific antigenic determinants [8]. The Cap protein is the major immunogenic structural protein of PCV2 and is responsible for neutralizing antibodies and inducing humoral and cellular immunity [9,10,11]. Therefore, the Cap protein is an important target in the design of a novel PCV2 vaccine using genetic engineering technology. Studies have shown that cytokines and multiple epitopes can enhance the immune effect of vaccines [12,13]. CD154 is a kind of II type transmembrane protein. The combination of CD154 and CD40 can promote the proliferation of activated, B lymphocytes, T lymphocytes, natural killer cells, and antibody production [14]. Granulocyte macrophage colony-stimulating factor (GM-CSF) has been used as a vaccine adjuvant, which enhances the immunogenicity of the vaccine by adjusting the number and maturity of local dendritic cells [15,16]. T and B cell epitopes enhance the immune response through the international with T cell receptors (TCR) and B cell receptors (BCR), respectively [17]. Early experiments conducted by Yunyan Wu in our laboratory confirmed that the combination of the dominant epitopes of T and B cells with Cap can enhance the immunogenicity of the vaccine. The PCV2 commercial vaccine selected in this experiment is a subunit vaccine with Cap as the antigen, which can indirectly serve as a control group without the addition of the dominant epitope of TB cells.

In the present study, the recombinant fusion proteins of TB-Cap, TB-Cap-CD154 and TB-Cap-GM-CSF were expressed in mammalian expression systems and the recombinant fusion proteins were used to further evaluate its vaccine potential in terms of immunogenicity and protection against PCV2 challenge in BALB/c mice.

## 2. Materials and Methods

### 2.1. Cells Lines, Plasmids and Viruses

Recombinant lentiviruses were packaged in human embryonic kidney 293 (HEK-293T) cell line. Chinese hamster ovarian cells K1 (CHO-K1) cells were infected with recombinant lentivirus [3]. PCV2 was propagated in porcine kidney 15 cells (PK15) [5]. Lentivirus vector pCDH-CMV-EF1-Puro, packaging plasmid PLP, PLP1 and PLP2 were preserved by the Microbiology Laboratory of South China Agricultural University (Fitgene Biological Technology Co., Ltd., Guangzhou, China). Codon-optimized genes encoding Cap protein (GenBank No. HM038034.1) of the PCV2 LG strain, porcine CD154 (GenBank No. NM_214126.1) and porcine GM-CSF (GenBank No. NM_214118.2) were synthesized and cloned into pUC57 plasmid (Bioengineering Co., Ltd., Shanghai, China). PCV2 commercialized subunit vaccine (Yibang Biological Engineering Co., Ltd., Qingdao, China) and PCV2 YZ strain (GenBank no. EU503040.1) were used in this study.

### 2.2. Construction of Recombinant Lentivirus Plasmids and Production of Lentivirus

Previous studies by Wu Yunyan et al. in our laboratory screened the specific the dominant epitopes of T and B cells and have confirmed that the combination of the dominant epitopes of T cell and B cell on Cap protein and Rep protein with Cap proteins can enhance immune-protective antibodies (Table 1). These parts are connected with a flexible linker (GGCGGC), which can provide structure flexibility, improve protein stability, or increase biological activity [18]. The pUC57-TB-Cap, pUC57-TB-Cap-CD154, pUC57-TB-Cap-GM-CSF and pCDH-CMV-EF1-Puro were digested with double enzymes (Thermo Fisher Scientific, Waltham, MA, USA), ligated using T4 ligase (Thermo Fisher Scientific, Waltham, MA, USA), and transformed into *E. coli* DH5α cells (Weidi Biotechnology Co., Ltd., Shanghai, China). The recombinant lentivirus plasmids used in this study and indicates the inserted gene fragments of B, T cell epitopes, Cap, CD154 and GM-CSF (Figure 1). The recombinant lentivirus plasmids, named as pCDH-TB-Cap, pCDH-TB-Cap-CD154 and pCDH-TB-Cap-GM-CSF, which were identified by polymerase chain reaction (PCR), restriction endonucleases digest and sequence analysis.

### 2.3. Infection CHO-K1 of Recombinant Lentivirus

Recombinant lentivirus plasmids pCDH-TB-Cap, pCDH-TB-Cap-CD154, pCDH-TB-Cap-GM-CSF and packaging plasmid transfected into HEK-293T cells by Lipofectamine™ 3000 Transfection Reagent (Thermo Fisher Scientific, Waltham, MA, USA). Briefly, 125 µL of reduced serum medium and 2.5 µg plasmid (PLP: PLP1: PLP2: recombinant plasmid = 1:1:1:1) were added to a sterile 1.5 mL centrifuge tube, followed by 5 µL Lipo 3000™ of liposomal transfection buffering reagent, and gently mixed before leaving at room temperature for 5 min. The prepared solution was added evenly to a 6-well cell culture plate (approximately 4 × 10^5^ cells). Three recombinant lentiviruses HIV-1-TB-Cap, HIV-1-TB-Cap-CD154 and HIV-1-TB-Cap-GM-CSF, respectively, were produced. For the recombinant lentivirus HIV-1-TB-Cap, HIV-1-TB-Cap-CD154, HIV-1-TB-Cap-GM-CSF and HIV-1-PCDH, 200 μL were added to CHO-K1 cells seeded in 6-well dishes, while a negative control was set. Then they were cultured at 37 °C with 5% CO_2_ for 24 h to enable the transfected plasmid to carry out gene integration with the CHO-K1 cells. 

The lentiviral vector carries the puromycin resistance gene, so the CHO-K1 cells that successfully expressed the recombinant protein have puromycin resistance. After the CHO-K1 cells returned to normal growth, the final concentration of puromycin was 14 µg/mL, and the puromycin was supplemented every 48 h. When all negative controls died, CHO-K1 recombinant cells successfully integrated with exogenous genes TB-Cap, TB-Cap-CD154 and TB-Cap-GM-CSF were obtained.

### 2.4. Detection of the Recombinant Proteins in CHO-K1 Cells

The recombinant cells were harvested and lysed. Protein samples were separated by 12.5% SDS-PAGE and transferred to polyvinylidene fluoride membrane. The transferred membranes were blocked by incubating with 5% skim milk in Tris-Buffered Saline Tween-20 (TBST) for 30 min and incubated overnight at 4 C with the anti-PCV2 positve serum primary antibodies at 1:2000 dilution. HRP-labeled goat anti-porcine antibody (Beyotime Biotechnology, Shanghai, China) was used as the secondary antibody. Protein bands were detected using ECL chemiluminescence system and analyzed by ImageJ-2.2.0 Launcher software. To further verify whether the recombinant protein is successfully expressed. The TB-Cap, TB-Cap-CD154 and TB-Cap-GM-CSF fusion proteins of CHO-K1 recombinant cells were identified by IFA. The cell culture plates were washed three times with PBS and fixed with cold absolute ethanol for 30 min on ice. After washing three times with PBS, cells were incubated with mouse anti-His (Beyotime Biotechnology, Shanghai, China) or anti-PCV2 positive serum antibody at 37 °C for 1 h. The cells were washed three times and stained with FITC-labeled goat anti-mouse IgG (Boshide, NY, USA) or FITC-labeled goat anti-porcine (Beyotime Biotechnology, Shanghai, China) at 37 °C for 1 h. After washing three times with PBS, the cells were observed under fluorescence microscope. 

### 2.5. Animals, Vaccination, and Challenge Experiments

The cell pellets were lysed by ultrasonication in an ice-water bath to harvest the recombinant protein. Serial dilutions of His standard protein were used for Western blot to generate the standard curve as a quantitative reference. The recombination protein was emulsified with ISA 201VG adjuvant at a ratio of 1:1 (w/o/w) to prepare a subunit vaccine. Forty BALB/c mice were purchased from Southern Medical University. Animal experiments were supported by the Animal Ethics Committee of the South China Agricultural University. Mice were randomly divided into five groups and every group contained eight mice. Mice in group A (negative control group) were vaccinated with 200 µL PBS. Mice in group B (positive control group) were immunized subcutaneously with 200 µL commercial PCV2 subunit vaccine. Mice in groups C–E were immunized subcutaneously with 50 µg TB-Cap, TB-Cap-CD154 and TB-Cap-GM-CSF subunit vaccines. Booster immunizations with the same dose of the vaccine were administered 14 and 28 days after the first vaccination. At 42 days post-primary immunization, all groups were intraperitoneally challenged with 1 × 10^5.7^ TCID50 PCV2.

### 2.6. Serology

Serum of four mice from each group were randomly collected on days 0, 7, 14, 21, 28, 35 and 42 after the first vaccination. Then, the separated serum was used to test the PCV2-specific antibody level according to the manufacturer’s instruction of Antibody Test Kit for PCV2 (Fender Biotechnology Co., Ltd., Shenzhen, China). Antibody levels were presented as the values of optical density (OD450 nm).

### 2.7. Lymphocyte Proliferation Assay

On day 45 after the first vaccination, three mice were randomly selected from each group to isolate spleen lymphocytes. Lymphocytes were resuspended at 5 × 106 cells/mL in RPMI-1640 supplemented with 8% FBS. Lymphocytes were seeded in 96-well flat-bottom plates at 100 µL per well and stimulated at 37 °C for 48 h with Concanavalin A (Soleibao Technology Co., Ltd., Beijing, China) as positive control, PCV2 as the stimulant and RPMI-1640 as negative control. Then, 10 µL Cell Counting Kit-8(CCK-8) (Yishan Biological Technology Co., Ltd., Shanghai, China) was added into each well and incubated at 37 °C incubator for 4 h. The reaction was terminated by adding 100 µL of DMSO. The stimulation index (SI) = (mean OD450 nm of PCV2 stimulated cells − mean OD450 nm of negative control cells)/(mean OD450 nm of positive control cells − mean OD450 nm of negative control cells).

### 2.8. Detection of CD4+ and CD8+ by Flow Cytometry

APC mouse anti-CD3e monoclonal antibody, FITC mouse anti-CD4a monoclonal antibody and PE mouse anti-CD8a monoclonal antibody (Thermo Fisher Scientific, Waltham, MA, USA) were added to lymphocytes. The reaction was carried out in the dark at 4 °C for 30 min. The cells were washed with 1.5 mL of flow cytometry staining buffer. The cells were resuspended with 500 µL flow cytometry buffer and analyzed by flow cytometry. The percentages of CD3, CD4+ and CD8+ and the CD4+/CD8+ ratio of cells were analyzed.

### 2.9. Detection of the Viremia by PCR

Serum samples were collected after day 28 of challenge. The viral DNA were extracted using a universal E.Z.N.A.^®^ Viral DNA Kit (Feiyang Biological Engineering Co., Ltd., Guangzhou, China) according to the manufacturer’s instructions. Presence of PCV2 DNA was determined using PCR with reaction mixture 1 μL DNA, 20 pmol of each primer (PCV2-Cap-F: GACGGG-TATCACGGAGAAGAG and PCV2-Cap-R: ACAGCCTGGTGTTGAAGATGC) and 22 μL of MIX PCR (Tsingke Biotechnology Co., Ltd., Guangzhou, China). PCR was performed using the following conditions: initial denaturation at 98 °C for 2 min; 30 cycles of 98 °C for 10 s, 60 °C for 30 s, and 72 °C for 10 s; final extension of 72 °C for 2 min.

### 2.10. Real-Time PCR

The concentrations of pCDH-TB-Cap standard plasmid were determined by Nanodrop 2000. The standard plasmid was diluted to 101~108 copies/mL to establish a standard curve. Serum, lungs and kidneys were collected 28 days after challenge for viral nucleic acid detection by qRT-PCR. The reaction system was 20 µL, consisting of 10 µL of 2 × ChamQ Universal SYBR qPCR Master Mix (Vazyme, Nanjing, China), 0.4 µL (20 pmol) of each primer (PCV2-Cap-F: GACGGGTATCACGGAGAAGAG and PCV2-Cap-R: ACAGCCTGGTGTTGAAGATGC), 1 µL of DNA template and 8.2 µL of RNase-free water. The reaction conditions were 95 °C for 30 s, 95 °C for 10 s, 60 °C for 15 s, for a total of 40 cycles. Additionally, an analytical curve was obtained by using the IQ5 RT-PCR Detection System under the reaction conditions. The viral load of the sample was calculated based on the standard curve and the analytical curve.

### 2.11. IL-2, IFN-Gamma, IL-4 and IL-10 Cytokine Detection

Serum samples of mice in each group were collected on the 28th day after challenge. The total RNA was extracted using a universal E.Z.N.A.^®^ Total RNA Kit (Feiyang Biological Engineering Co., Ltd., Guangzhou, China) according to the manufacturer’s instructions. Then, reverse transcription was performed on 1000 ng of total RNA using the HiScript III RT SuperMix for qPCR (Vazyme, Nanjing, China). The transcription levels of cytokines IL-2, IFN-gamma, IL-4 and IL-10 were detected by relative quantitative method. Primer sequence information is shown in Table 2.

### 2.12. Statistical Analysis

All analyses were carried out using GraphPad Prism software. Two-way ANOVA analysis was used for statistical analyses between multiple groups. *p*-value less than 0.05 was considered statistically significant.

## 3. Results

### 3.1. Dentification of Recombinant Lentivirus Plasmids 

The recombinant lentivirus plasmids were dentification by PCR, restriction enzyme with Nhe I and BamH I/Not I and sequencing. Three genes of TB-Cap, TB-Cap-CD154 and TB-Cap-GM-CSF were successfully amplified with 1143, 1931 and 1578 bp sizes. Two fragments, respectively, were observed after three recombinant plasmids were digested by Nhe I and BamH I/Not I. Then, the correct lentivirus plasmids were chosen by sequencing. The results showed that the recombinant lentiviral plasmids were successfully constructed and named pCDH-TB-Cap, pCDH-TB-Cap-CD154 and pCDH-TB-Cap-GM-CSF (Figure 2).

### 3.2. Expression of the Recombinant Proteins

The plasmid and lentiviral packaging plasmids were co-transfected to HEK-293T cells and produced recombinant lentivirus in HEK-293T cell. CHO-K1 cells were infected with recombinant lentivirus HIV-1-TB-Cap, HIV-1-TB-Cap-CD154, HIV-1-TB-Cap-GM-CSF and HIV-1-pCDH, then cultured for 24 h and screened with 14 µg/mL puromycin for 8 days. Western blot was used to detect whether CHO-K1 cells successfully expressed the fusion protein. Western blot showed that the Cap and His proteins were detected in recombinant CHO-K1 cells. Bands corresponding to the Cap and His proteins were not detected in uninfected cells or infected with HIV-1-pCDH. Consistently, indirect IFA showed that recombinant lentivirus infected CHO-K1 cells could be stained with anti-PCV2 serum and anti-His mAbs. This green fluorescence was not observed in cells belonging to the negative control group. These results confirmed that the fusion proteins TB-Cap, TB-Cap-CD154 and TB-Cap-GM-CSF were successfully expressed in the cells infected with the recombinant lentivirus (Figure 3).

### 3.3. Screening and Stability Identification of Monoclonal Cells

The recombinant monoclonal CHO-K1-TB-Cap, CHO-K1-TB-Cap-CD154 and CHO-K1-TB-Cap-GM-CSF cell lines with the highest relative expression were screened using 96-well plate via limited dilution. The recombinant monoclonal cells were cultured continuously for 25 generations, and the same number of cells (4 × 10^5^ cells) were taken every 5 generations to test the stability of the fusion protein. The stability of fusion proteins in recombinant CHO-K1 cells were evaluated using anti-PCV2 positive serum antibody. Western blot results showed that the 5th, 10th, 15th, 20th and 25th generations of the recombinant CHO-K1 cells were stably expressed fusion proteins (Figure 4).

### 3.4. Humoral Immune Response

The TB-Cap, TB-Cap-GM-CSF and TB-Cap-CD154 fusion proteins emulsified with ISA 201VG adjuvant, respectively. The 5-week-old BALB/c mice were immunized with the same dose on the days 14 and 28 after the first immunization. The serum specific antibody levels were detected by indirect ELISA kit on the days 7, 14, 21, 28, 35 and 42 after the first immunization to evaluate immunogenicity of vaccines. The cutoff value of the S/P ratio was 0.2. The levels of anti-PCV2 antibodies in the vaccinated groups gradually increased with booster immunization. Animals in all the vaccinated groups, except those in the PBS control group and recombinant subunit vaccine group, were seropositive on day 21 after the primary immunization. On the days 21, 28 and 35 after the first immunization, the levels of anti-PCV2 antibodies in TB-Cap, TB-Cap-CD154 and TB-Cap-GM-CSF groups were significantly higher than that in the commercial vaccine group and PBS group (*p* < 0.0001). On the days 21, 28, 35 and 42 after the first immunization, the levels of anti-PCV2 antibodies in TB-Cap-GM-CSF and TB-Cap-CD154 groups were significantly higher than that in the TB-Cap group (*p* < 0.0001). These results showed that the three fusion proteins in this study could induce good humoral immune responses in mice and are better than the commercial vaccine group. The introduction of cellular molecules GM-CSF and CD154 could enhance the humoral immune response in mice (Figure 5).

### 3.5. Lymphocyte Proliferative Activity

Lymphocyte proliferation level is one of the most significant indices of cellular immune response. Lymphocyte proliferative activities of different vaccinated mice were determined with the CCK-8 method on day 42 after the first immunization. The lymphocyte proliferative level of TB-Cap, TB-Cap-CD154 and TB-Cap-GM-CSF groups was significantly higher than PBS group (*p* < 0.0001). The lymphocyte proliferative level of TB-Cap-CD154 and TB-Cap-GM-CSF groups was significantly higher than those in other groups (*p* < 0.05) (Figure 6). These results indicated that the TB-Cap, TB-Cap-CD154 and TB-Cap-GM-CSF fusion proteins could induce cellular immunity response in mice. The introduction of CD154 and GM-CSF molecules could enhance the cellular immune response of TB-Cap.

### 3.6. Detection of CD4+ T and CD8+ T Analysis by Flow Cytometry

In order to determine if the immune groups could cause changes in T cell subsets, we collected PBMC cells from the immunized mice and analyzed them by flow cytometry. The CD4+ T/CD8+ T ratio in each vaccine group was within the normal range. The ratio of CD4+ T cells and CD8+ T cells in mice immunized with TB-Cap, TB-Cap-CD154 and TB-Cap-GM-CSF were significantly higher than those of PBS and recombinant subunit vaccines (*p* < 0.01). Moreover, the ratio of CD4+ T/CD8+ T in TB-Cap-CD154 and TB-Cap-GM-CSF groups was significantly higher than that in other groups (Figure 7) (Table 3).

### 3.7. Immune Protection against PCV2 Challenge

The viremia of the serum was detected by PCR on day 28 after the challenge. The results showed that the positive rate of viremia was 100% in the PBS group and 20% in the recombinant subunit vaccine group. No viremia was detected in TB-Cap, TB-Cap-CD154 and TB-Cap-GM-CSF groups (Table 4).

Viral loads of the serum, lung and kidney were detected by real-time quantitative PCR on day 28 after the challenge. In serum and lung, the viral loads of TB-Cap, TB-Cap-CD154 and TB-Cap-GM-CSF groups were significantly lower than that of commercial vaccine and PBS groups (*p* < 0.0001), and the reduction effect was most obvious in serum. Data showed that in the kidney, the TB-Cap-CD154 and TB-Cap-GM-CSF groups had significantly lower viral loads than the other groups (*p* < 0.0001). These results indicated that the three fusion proteins prepared in this study could provide effective immune protection to mice and were superior to the recombinant subunit vaccine (Figure 8).

### 3.8. Analysis of IL-2, IFN-Gamma, IL-4 and IL-10

The levels of cytokine secretion are an important indicator of cellular immunity. The transcription levels of cytokines IL-2, IFN-gamma, IL-4 and IL-10 in serum were measured 28 days after challenge, and Th1 and Th2 responses of immunized mice were evaluated from the side. The transcription levels of cytokines in the TB-Cap, TB-Cap-CD154 and TB-Cap-GM-CSF groups were significantly higher than those in the PBS and recombinant vaccine groups (*p* < 0.0001). Moreover, the transcription level of cytokines in the TB-Cap-CD154 and TB-Cap-GM-CSF groups were higher than TB-Cap groups (*p* < 0.01). These results indicate that the TB-Cap protein fused with CD154 or GM-CSF factor can increase the transcription level of cytokines, reflecting the enhancement of the body’s cellular immune response from the side (Figure 9).

## 4. Discussion

PCVAD is a contagious disease of swine caused by PCV2. PCV2 infection can lead to immunosuppression of the host, which is complicated or secondary to other pathogen infections [19]. It is very harmful to the pig industry. The prevention and control of PCVAD are based on proper immunization and management practices [20]. Subunit vaccines are widely used because of their high safety and low side effects but their immunogenicity is low [21]. A number of strategies have been used to augment the potency of the immune response. Fusion expression of cytokines and immunogenic proteins has been shown to enhance the cellular immune response, the humoral immune response, or both [22]. The Cap protein encoded by ORF2 gene is the only structural protein of PCV2, and it is an ideal antigen for the development of PCV2 subunit vaccines [23,24]. CD154 is a member of the tumor necrosis factor (TNF) superfamily, which is primarily expressed by activated T cells [25,26]. CD40L interacts with CD40 and can stimulate the proliferation of activated B lymphocytes, T lymphocytes, natural killer cells and antibody production [27]. GM-CSF can activate antigen presenting cells (APCs) such as dendritic cells (DCs) or macrophages, which can regulate the number and maturity of local dendritic cells to enhance the immunogenicity of the vaccine [28]. Harcourt Jennifer L and Babai reported that use of CD154 and GM-CSF as an adjuvant in subunit vaccine enhanced humoral immune response against the vaccine [29,30]. It is indicated that CD154 and GM-CSF can be used as molecular adjuvants to significantly improve the immune protection of vaccines. In addition, the protective antigen gene has a synergistic effect with epitopes gene, which can enhance humoral and cellular immune responses in the body [31]. Thus, the present study aimed to assess whether CD154 and GM-CSF could improve immune responses against PCV2.

In this study, porcine CD154 and GM-CSF were applied as molecular adjuvants and fused with Cap protein, respectively. The fusion proteins TB-Cap, TB-Cap-CD154 and TB-Cap-GM-CSF were expressed by mammalian expression system and expression of these fusion proteins was confirmed using Western blotting and immunofluorescence analysis. To confirm the protective immune response induced by different fusion proteins, the mice were immunized and challenged with PCV2. The PCV2 commercial vaccine selected in this experiment is a subunit vaccine with Cap as the antigen, which can indirectly serve as a control group without the addition of the dominant epitope of TB cells. Lentiviral vectors are regarded as ideal retroviral vectors that can transfer target genes because of their many advantages such as a wide range of transfected cells, high infection efficiency, large volume of introduced gene fragments and stable expression, and are widely used in vaccine and drug research and development [32]. In order to ensure biological safety and prevent the production of active viruses during the construction of the lentiviral expression vector, this experiment uses a four-plasmid lentiviral expression vector. The lentiviral vector was divided into four originals, and the original expressing the foreign gene carried the puromycin resistance gene, which was used for subsequent screening of CHO-K1 cells that successfully expressed the foreign protein. The four plasmids are co-transfected into cells to assemble lentiviral vector particles that have lost the ability to replicate, thereby ensuring biological safety. Compared with other expression systems, the CHO-K1 expression system has many advantages, such as accurate post-transcriptional modification function, the expressed protein is closest to the native protein molecule in terms of molecular structure, physicochemical properties and biological function [33]. The fusion proteins TB-Cap, TB-Cap-CD154 and TB-Cap-GM-CSF expressed by CHO-K1 cells used in this experiment have better biological activity than other experiments. However, the disadvantage of the mammalian expression system is the low expression level. In order to increase the expression of foreign proteins, the recombinant CHO-K1 cells were cloned in this experiment and the one with the highest expression was selected, which can still maintain a stable and high expression level during the passage. In addition, this experiment trained monoclonal cells in suspension to further increase the expression level of exogenous proteins by increasing the number of cells.

The protective effect of animal immunity is the criterion for evaluating vaccines. The results of animal experiments showed that PCV2-specific antibodies could be produced in mice immunized with the three fusion proteins (*p* < 0.001), and TB-Cap-CD154 and TB-Cap-GM-CSF groups could induce higher humoral immune responses in mice (*p* < 0.001). The introduction of cellular molecules GM-CSF and CD154 could enhance the humoral immune response in mice. Lymphocyte proliferation level is one of the most significant indices of cellular immune response. The lymphocyte proliferation level results showed that the lymphocyte proliferation levels of TB-Cap-CD154 and TB-Cap-GM-CSF groups were significantly higher than that of other groups (*p* < 0.05). Additionally, the results of flow cytometry show that the three fusion proteins can effectively enhance the immune function of the body. Thus, these results showed that the fusion proteins TB-Cap, TB-Cap-CD154 and TB-Cap-GM-CSF prepared in this study have good immunogenicity, and the introduction of CD154 and GM-CSF can induce the body to produce earlier humoral immune response. Further, after PCV2 challenge, no PCV2 viremia was detected in the vaccinated groups. Viral loads in serum, lung and kidney samples of the all vaccinated groups were significantly lower than the PBS group (*p* < 0.001). The secretion levels of IL-2 and IL-10 cytokines in the TB-Cap, TB-Cap-CD154 and TB-Cap-GM-CSF groups were significantly higher than those in the PBS and recombinant vaccine groups (*p* < 0.0001). Moreover, cytokines IL-2, IFN-gamma, IL-4 and IL-10 were elevated when introduced to CD154 and GM-CSF. These results indicate that TB-Cap, TB-Cap-CD154 and TB-Cap-GM-CSF could provide immune protection against PCV2 infection in mice. In addition, the fusion proteins in this study should be tested in pigs. It is hoped that CD154 and GM-CSF molecules will be beneficial to improve immune responses and protective efficacy of the vaccine against PCV2.

In conclusion, three fusion proteins (TB-Cap, TB-Cap-CD154 and TB-Cap-GM-CSF) were expressed by the mammalian expression system in this study, and stable CHO-K1 cell lines expressing the fusion proteins were successfully established. These results indicated that CD154 and GM-CSF is a powerful immune adjuvant for PCV2 subunit vaccines to enhance humoral immune response and could protect mice against the PCV2 challenge. Thus, three fusion proteins TB-Cap, TB-Cap-CD154 and TB-Cap-GM-CSF can be used as the novel PCV2 subunit vaccine candidate for control of PCV2. Now, the relative works are under way in pigs and may help circumvent some of the anatomical and physiological differences between mice and pigs.

## Figures and Tables

**Figure 1 vetsci-08-00211-f001:**
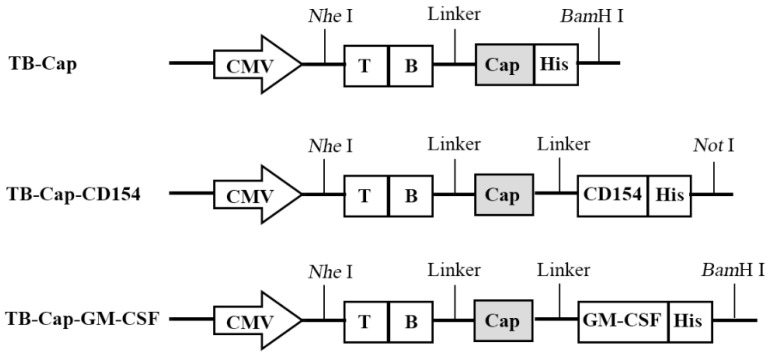
Schematic diagram of different recombinant genes. In order to improve the immunogenicity of Cap protein, recombinant genes TB-Cap, TB-Cap-CD154 and TB-Cap-GM-CSF were designed.

**Figure 2 vetsci-08-00211-f002:**
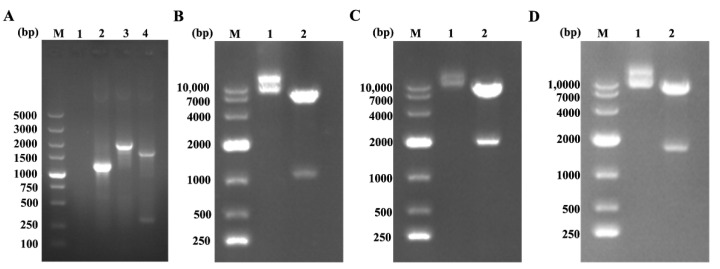
The recombinant plasmids were identified by PCR and restriction enzyme with Nhe I and BamH I/Not I. (**A**) The recombinant plasmids were identified by PCR. M: 5K DNA Marker; Lane 1: negative control; Lane 2: TB-Cap; Lane 3: TB-Cap-CD154; Lane 4: TB-Cap-GM-CSF; (**B**) pCDH-TB-Cap digestion products. M: 10K DNA Marker; Lane 1: pCDH-TB-Cap plasmid; Lane 2: pCDH-TB- Cap digestion products; (**C**) pCDH-TB-Cap-CD154 digestion products. M: 10K DNA Marker; Lane 1: pCDH-TB-Cap-CD154 plasmid; Lane 2: pCDH-TB-Cap-CD154 digestion products; (**D**) pCDH-TB-Cap-GM-CSF digestion products. M: 10K DNA Marker; Lane 1: pCDH-TB-Cap-GM-CSF plasmid; Lane 2: pCDH-TB-Cap-GM-CSF digestion products.

**Figure 3 vetsci-08-00211-f003:**
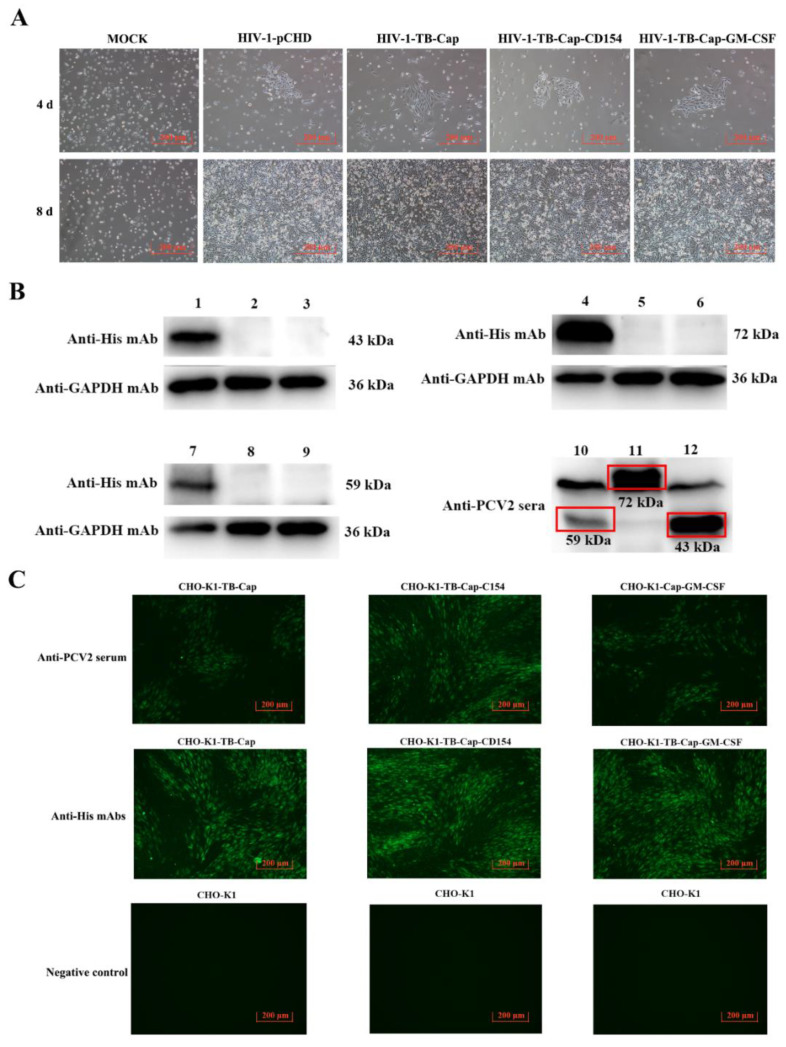
The recombinant CHO-K1 cells were screened with puromycin and the expression of recombinant TB-Cap, TB-Cap-CD154 and TB-Cap-GM-CSF proteins were analyzed by Western blotting and indirect immuno-fluorescence. (**A**) CHO-K1 cells were infected with different lentiviruses and screened by adding 14 µg/mL puromycin. All cells not infected with the recombinant lentivirus died after 8 days; (**B**) the expression of recombinant TB-Cap, TB-Cap-CD154 and TB-Cap-GM-CSF were analyzed by Western blotting. Lanes 2,5,8: cell lysates of CHO-K1 cells (negative control); Lanes 3,6,9: cell lysates of CHO-K1-pCDH cells; Lane 1: cell lysates of CHO-K1-TB-Cap; Lane 4: cell lysates of CHO-K1-TB-CD154; Lane 7: cell lysates of CHO-K1-TB-Cap-GM-CSF; Lane 10: CHO-K1-TB-Cap-GM-CSF; Lane 11: cell lysates of CHO-K1-TB-CD154; Lane 12: cell lysates of CHO-K1-TB-Cap; (**C**) the fusion proteins expressions of recombinant TB-Cap, TB-Cap-CD154 and TB-Cap-GM-CSF were analyzed by indirect immunofluorescence, using Anti-PCV2 positive serum antibodies and His monoclonal antibody as primary antibodies, respectively.

**Figure 4 vetsci-08-00211-f004:**
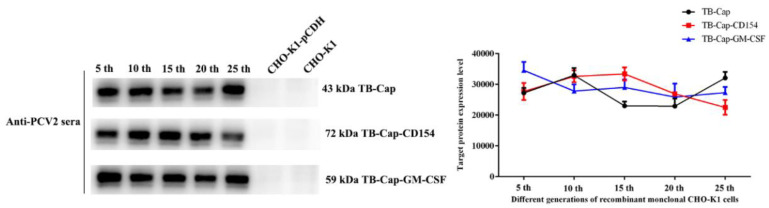
Stability identification of monoclonal cells. Stability identification of monoclonal cells lines. Western blot was used to analyze the fusion protein expression of the 5th, 10th, 15th, 20th and 25th generations of monoclonal cell lines, and Image J was used to quantify the protein gray value. All data shown represent the mean ± SD calculated from three independent experiments.

**Figure 5 vetsci-08-00211-f005:**
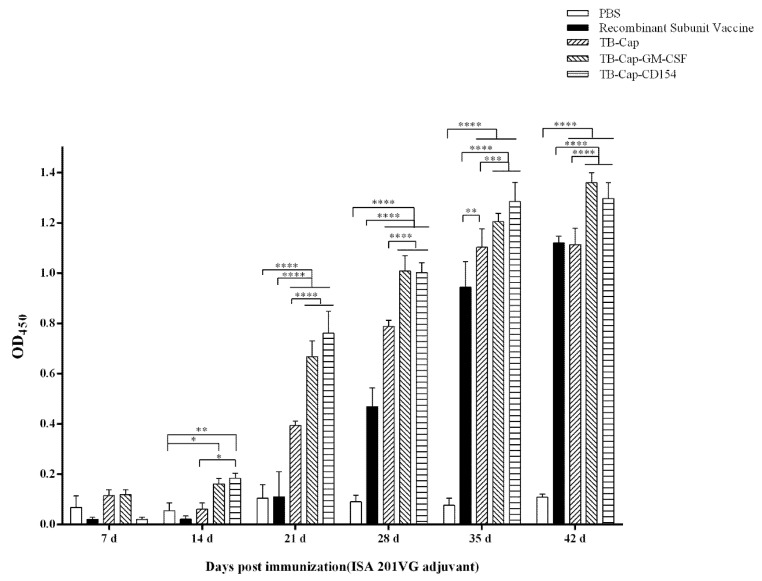
PCV2-specific humoral immune responses in different groups. All mice in the immunization groups were immunized again on the days 14 and 28 after the first immunization. Serum samples were collected on days 0, 7, 14, 21, 35, and 42 after first immunization and the serum antibody levels were detected by indirect ELISA. The cutoff value of S/P ratio is 0.2. d: day. Data represent the mean ± SD. An asterisk (*) shows a statistically significant difference between the indicated groups (*p* < 0.05). (*): *p* < 0.05; (**): *p* < 0.01; (***): *p* < 0.001; (****): *p* < 0.0001.

**Figure 6 vetsci-08-00211-f006:**
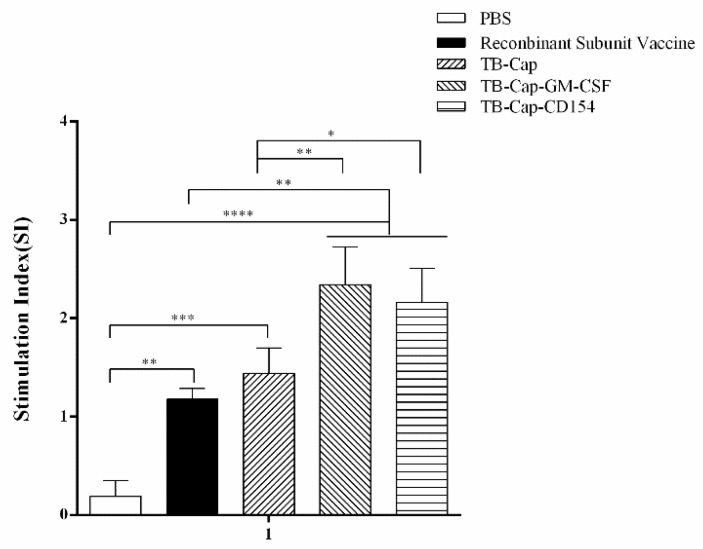
Lymphocyte proliferation levels in different groups. On day 42 after the first immunization, lymphocytes were collected and the proliferation was detected by CCK-8. The stimulation index (SI) = (mean OD_450 nm_ of PCV2 stimulated cells − mean OD_450 nm_ of negative control cells)/(mean OD_450 nm_ of positive control cells− mean OD_450 nm_ of negative control cells). Data represent the mean ± SD. An asterisk (*) shows a statistically significant difference between the indicated groups (*p* < 0.05). (*): *p* < 0.05; (**): *p* < 0.01; (***): *p* < 0.001; (****): *p* < 0.0001.

**Figure 7 vetsci-08-00211-f007:**
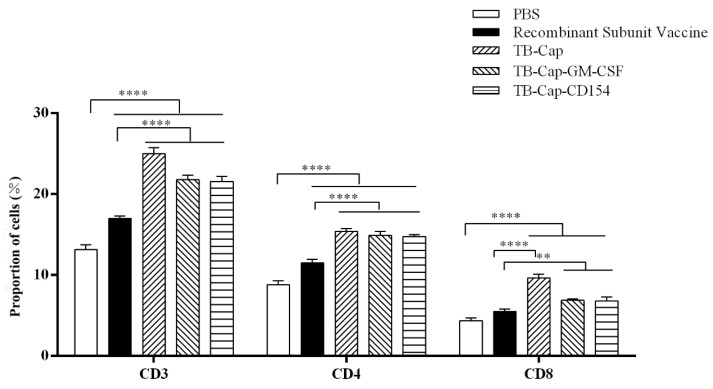
The changes of lymphocyte subsets after immunization were analyzed by flow cytometry. On day 42 after the first immunization, lymphocytes were collected and CD4+ T cells and CD8+ T cells were analyzed by flow cytometry. Data represent the mean ± SD. An asterisk (*) shows a statistically significant difference between the indicated groups (*p* < 0.05). (**): *p* < 0.01; (****): *p* < 0.0001.

**Figure 8 vetsci-08-00211-f008:**
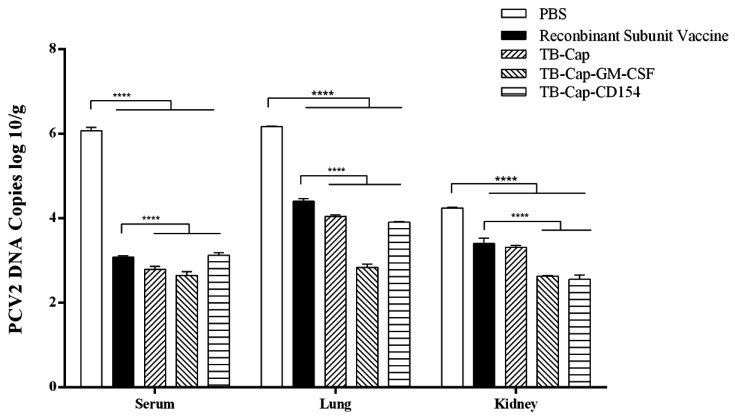
Detection of PCV2 viral load after challenge. On the day 42 after the first immunization, all mice in the immunization group were challenged with 1×10^5.7^ TCID_50_ PCV2. Serum, lungs and kidneys were collected and viral load was detected by real-time quantitative PCR on the day 28 after challenge. Data represent the mean ± SD. An asterisk (*) shows a statistically significant difference between the indicated groups (*p* < 0.05). (****): *p* < 0.0001.

**Figure 9 vetsci-08-00211-f009:**
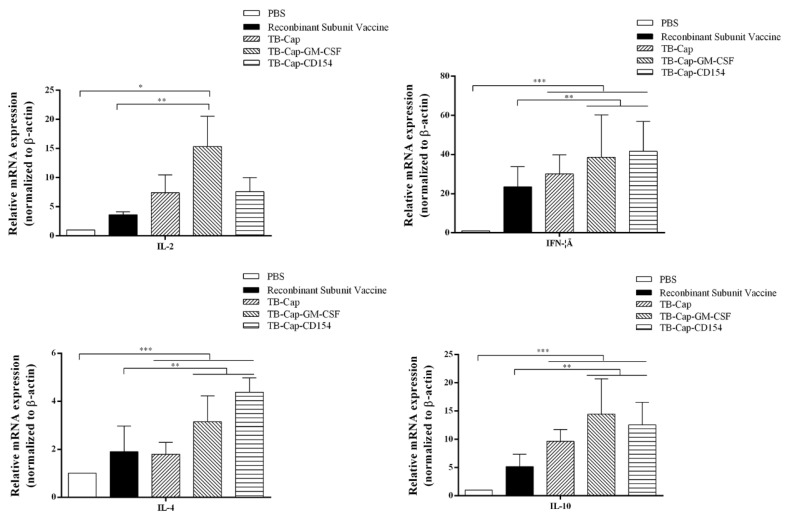
The levels of IL-2, IFN-gamma, IL-4 and IL-10 cytokine secretion. The mRNA levels of IL-2, IFN-gamma, IL-4 and IL-10 were analyzed by qRT-PCR in serum of mice on the day 28 after challenge. Data represent the mean ± SD. An asterisk (*) shows a statistically significant difference between the indicated groups (*p* < 0.05). (*): *p* < 0.05; (**): *p* < 0.01; (***): *p* < 0.001.

**Table 1 vetsci-08-00211-t001:** B, T cell linear epitopes specific information.

Epitopes	At the Cap/Rep Protein Position	Amino Acid Sequence
B 1	C61~85	TVRTPSWAVDMMRFNIDDFVPPGGG
B 2	C133~131	QGDRGVGSTAVILDDNFVT
B 3	C169~180	STIDYFQPNNKR
B 4	C192~202	NVDHVGLGTAF
R 1	R81~100	CHIEKAKGTDQQNKEYCSKE
R 2	R201~220	KWWDGYHGEEVVVZDDFYGW

**Table 2 vetsci-08-00211-t002:** Primers for qRT-PCR amplification used in this study.

Gene	Sequence (5′–3′)	Size (bp)
β-actin	F: TGCTGTCCCTGTATGCCTCTG	100
	R: CTTTGATGTCACGCACGATTTC	
IL-2	F: GCTCTACAGCGGAAGCACAG	382
	R: CATCTCCTCAGAAAGTCCACCAC	
IFN-gamma	F: CTCAAGTGGCATAGATGTGGAAG	268
	R: CTGGACCTGTGGGTTGTTGAC	
IL-4	F: GTCATCCTGCTCTTCTTTCTCG	379
	R: TGATGCTCTTTAGGCTTTCCAG	
IL-10	F: ACAACATACTGCTAACCGACTCC	296
	R: TTCATTCATGGCCTTGTAGACAC	

**Table 3 vetsci-08-00211-t003:** Analysis of CD4+ and CD8+ of immunized mice by flow cytometry.

Group	CD3 (%)	CD4+ (%)	CD8+ (%)	CD4+/CD8+ Ratio
PBS	13.17 (±0.34)	8.81 (±0.28)	4.36 (±0.21)	2.02 (±0.03)
Recombinant subunit vaccine	16.99 (±0.17)	11.50 (±0.27)	5.49 (±0.17)	2.09 (±0.01)
TB-Cap	25.02 (±0.43)	15.38 (±0.21)	9.64 (±0.26)	1.60 (±0.03)
TB-Cap-CD154	21.56 (±0.37)	14.76 (±0.13)	6.80 (±0.29)	2.17 (±0.07)
TB-Cap-GM-CSF	21.78 (±0.33)	14.9 (±0.29)	6.88 (±0.09)	2.17 (±0.01)

**Table 4 vetsci-08-00211-t004:** The viremia of different immune groups.

Group	Day 14 after the PCV2 Challenge						Day 28 after the PCV2 Challenge					
	1	2	3	4	5	Positive rate	1	2	3	4	5	Positive rate
PBS	+	+	+	+	+	100	+	+	+	+	+	100
Recombinant subunit vaccine	−	−	−	+	−	20	−	−	−	+	−	20
TB-Cap	−	−	−	−	−	0	−	−	−	−	−	0
TB-Cap-CD154	−	−	−	−	−	0	−	−	−	−	−	0
TB-Cap-GM-CSF	−	−	−	−	−	0	−	−	−	−	−	0

## Data Availability

The data presented in this study are available on request from the corresponding author.

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
