# Peer review of "Fusion Expression and Immune Effect of PCV2 Cap Protein Tandem Multiantigen Epitopes with CD154/GM-CSF"

_vetsci, 2021, doi:10.3390/vetsci8100211_

Round 1

Reviewer 1 Report

Porcine circovirus type 2 (PCV2) is a very important pathogenic causing porcine circovirus-associated diseases (PCVAD). In order to effectively prevent and control PCVAD, the authors attempted to develop a new safe, efficient and cheap subunit vaccine, as described in the current manuscript. The idea is to fuse two potential immunoadjuvants (CD154 or GM-CSF) with Cap, aiming for an enhanced immunogenicity of the vaccine. Mice experiments showed that, the new subunit vaccine candidates could induce a high level of cellular immune response, decrease the viral loads in serum, lung and kidney, and upregulate the levels of IL-2, IL-4, IL-gamma, and IL10. Overall, the current paper shares a similar framework of experiment designs with a paper published in 2013 in Vaccines “Enhanced protective immune response to PCV2 subunit vaccine by co-administration of recombinant porcine IFN-gamma in mice”. Moreover, instead of overlaying results from different experiments, a clear description, a deep analysis and a better presentation/interpretation of results are needed, so that the readers could follow the flow of the paper easily.

  1. In the Introduction part, it is not clear why there is a need to develop a new subunit vaccine. What are the drawbacks of the existing subunit vaccines?
  2. The structure of Material and Methods part needs to be reorganized.

For example: 2.2 “Construction of recombinant lentivirus plasmids” could be changed to “Construction of recombinant lentivirus plasmids and production of lentivirus”; Then it would be logical to have 2.3 “Infection of CHO-K1 with lentivirus”.

Moreover, 2.4 “Western blot analysis of the protein expression” and 2.5 “Immunofluorescence assay” could be categorized to one section “Detection of the recombinant proteins in CHO-K1 cells”

  1. Part of Material and Methods needs to be written in detail.

For example, in Section 2.3, for the production of lentivirus, how many cells did you seed for transfection? How many plasmids did you use for transfection? What is the amount of lentivirus you used to transduce CHO-K1 cells?

In Section 2.6, how did you prepare the immunogens for mice immunization? How did you determine the concentration of proteins?

  1. In Section 2.7, it is not clear that whether the Antibody test kit for PCV2 determined PCV2-specific antibodies or neutralization antibodies? Would be interesting to check the levels of neutralization antibodies as well.
  2. The title of Section 2.11 “Real-time polymerase chain reaction” could be changed to “Real-time PCR”
  3. The two paragraphs below “3. Results” needs to be deleted. The first paragraph is not scientific content, and the second paragraph is a repetition of Section 3.1. The reviewer do suggest the authors to polish their manuscript before submission.
  4. Extensive editing of English language is a must.

For example, in Line 15 and 16, Cap is the principle immunogenic protein and stimulate the produce humoral and cellular immunity; in Line 25 and 26, And mice in all the vaccinated groups significantly higher levels than PBS group; in Line 391, CD40L interacts with CD40 can stimulate the proliferation of activated B lymphocytes”

  1. To increase the novelty of the current paper, would be interesting to study the effect of the new subunit vaccines against PCV2d strains. It is known that PCV2d outbreaks happen in the background of extensive vaccination. Would be of an added benefit if the new subunit vaccines could control PC2d infection. Like the authors stated in the last paragraph, the data would be more convincing and exciting if the pig model was used.

Author Response

Dear Reviewers:

Thank you very much for your comments concerning our manuscript. Those comments are all valuable and very helpful for revising and improving our paper, as well as the important guiding significance to our researches. We have studied comments carefully and have made correction which we hope meet with approval. Revised portion are marked in red in the paper. The main corrections in the paper and the responds to your comments are as flowing:

Response:

Q1: In recent years, the main types of PCV2 commercial vaccines on the market are inactivated vaccine and subunit vaccine [6]. Inactivated vaccines are antigenic but lacked the capability to stimulate the body to produce complete cellular immune responses. The current PCV2 subunit vaccines are often poorly immunogenic as they represent a small portion of the pathogen and may require multiple vaccinations with large doses to provide protective immunity. In order to effectively prevent and control PCVAD, there is an urgent to develop a new safe, efficient and cheap vaccine. Among them, the development of enhancing the immunogenicity of the PCV2 subunit vaccine is particularly important (lines 53-62).

Q2: Thank you for pointing out this problem, which is of great significance to the logic and structure of the manuscript. We reorganized part of the structure of materials and methods.

Q3: Thank you for your professional question. Recombinant lentivirus plasmids pCDH-TB-Cap, pCDH-TB-Cap-CD154, pCDH-TB-Cap-GM-CSF and packaging plasmid transfected into HEK293 cells by Lipofectamine™ 3000 Transfection Reagent (Thermo Fisher Scientific, USA). Briefly, add 125 µL of reduced serum medium and 2.5 µg plasmid (PLP: PLP1: PLP2: recombinant plasmid=1:1:1:1) to a sterile 1.5 ml centrifuge tube, followed by 5 µL Lipo 3000™ of liposomal transfection buffering reagent, and gently mix before leaving at room temperature for 5 min. Add the prepared solution evenly to a 6-well cell culture plate (approximately 4×105 cells). And produced 3 recombinant lentiviruses HIV-1-TB-Cap, HIV-1-TB-Cap-CD154 and HIV-1-TB-Cap-GM-CSF, respectively. The recombinant lentivirus HIV-1-TB-Cap, HIV-1-TB-Cap-CD154, HIV-1-TB-Cap-GM-CSF and HIV-1-pCDH were added 200 µL to CHO-K1 cells seeded in 6-well dishes, while a negative control was set (lines 121-129).

Q4: Your comments are very scientific and rigorous, and deserve our respect. The cell pellets were lysed by ultrasonication in an ice-water bath to harvest the recombinant protein. A serial dilutions of His standard protein were used for western blot to generate the standard curve as a quantitative reference. The recombination protein was emulsified with ISA 201VG adjuvant at a ratio of 1:1 (w/o/w) to prepare a subunit vaccine (lines 156-159).

Q5: I'm very sorry for the unclear description. The separated serum was used to test the PCV2-specific antibodies level according to the manufacturer's instruction of Antibody Test Kit for PCV2.

In addition, we revised and strengthened the discussion and English sentences. Once again, thank you very much for your comments and suggestions.

Best regards.

Your sincerely,
Mao Qian

Reviewer 2 Report

Congratulations to the Authors for this extensive work and manuscript. Certainly, the future use of the revised output may enhance Veterinary Medicine to tackle such a pressing issue for pig husbandry and production.

Please take the following suggestions to clarify some information/data that will improve the manuscript quality:

  • Manuscript should be proof-read by English native/professional writer (particular attention to grammar and phrase construction, as some errors prove the reading very challenging)
  • Line 43, add Virus classification in order to prepare for future M&M descriptions (RNA) and better context
  • Lines 45-48,PCVAD is not just one disease... several clinical presentations from the infection by PCV2. Please revise
  • Line 78, M&M are not comprehensible for the procedures should be clearly described. Very confusing the description in some subsections, e.g. what is the rational to use 2 cell lines? Was the transfection system (Lipofectamine) used in both cell lines? Any protocol difference in the transduction? Also (Line 97-98), one construct (PURO) is not further named, revise if still needed.
  • Line 120, Puromycin used and not explained (selection marker)
  • Line 132, confusing stepwise description on antibody incubation, please revise and clarify
  • Line 122-124, unnecessary description.
  • Line 223, Correct "Identification"
  • Figure 3A and Figure 3C, image magnification scale should be present
  • Line 271(3.3.), as clonal populations, the relative number of cells is not specified and WB image (Figure 4) is also lacking the internal control (host keeping gene expression fundamental). As similarly, in a previous image, Authors used GAPDH to ascertain the relative quantity of extracts and later on (Fig. 9) b-actin was used.
  • Line 391, correct references  [26,27] position in text
  • Line 379, Considering all the gathered information, Discussion is poor and not reflecting a comprehensive explanation of the sum of results obtained. Suggestion to revise and enhance Discussion and analysis of results.

Author Response

Dear Reviewers:

Thank you very much for your comments concerning our manuscript. Those comments are all valuable and very helpful for revising and improving our paper, as well as the important guiding significance to our researches. We have studied comments carefully and have made correction which we hope meet with approval. Revised portion are marked in red in the paper. The main corrections in the paper and the responds to your comments are as flowing:

Response:

Q1: We have added a virus classification to the manuscript. Porcine circovirus (PCV) is the smallest known animal viruses, belong to the genus Circovirus in the Circoviridae family and can be divided into three serotypes, Porcine circovirus type 1 (PCV1), Porcine circovirus type 2 (PCV2) and Porcine circovirus type 3 (PCV3) (lines 42-45).

Q2: Thank you for pointing out this problem. PCV2 infection in pigs causes a series of Porcine circovirus associated diseases (PCVAD), such as postweaning multisystemic wasting syndrome (PMWS), porcine dermatitis nephropathy syndrome (PDNS) and swine reproductive disorder syndrome (SRDS) and other varieties of clinical syndromes, which brings great economic losses to the global pig industry (lines 47-49).

Q3: I'm very sorry for the unclear description and your comments are very scientific and rigorous, and deserve our respect. Recombinant lentivirus plasmids pCDH-TB-Cap, pCDH-TB-Cap-CD154, pCDH-TB-Cap-GM-CSF and packaging plasmid transfected into HEK293 cells by Lipofectamine™ 3000 Transfection Reagent (Thermo Fisher Scientific, USA). Briefly, add 125 µL of reduced serum medium and 2.5 µg plasmid (PLP: PLP1: PLP2: recombinant plasmid=1:1:1:1) to a sterile 1.5 ml centrifuge tube, followed by 5 µL Lipo 3000™ of liposomal transfection buffering reagent, and gently mix before leaving at room temperature for 5 min. Add the prepared solution evenly to a 6-well cell culture plate (approximately 4×105 cells). And produced 3 recombinant lentiviruses HIV-1-TB-Cap, HIV-1-TB-Cap-CD154 and HIV-1-TB-Cap-GM-CSF, respectively. The recombinant lentivirus HIV-1-TB-Cap, HIV-1-TB-Cap-CD154, HIV-1-TB-Cap-GM-CSF and HIV-1-PCDH were added 200μL to CHO-K1 cells seeded in 6-well dishes, while a negative control was set (lines 119-129).

Transfection only occurs in HEK-293T cells. Because HEK-293T cells have higher transfection efficiency, they can be better packaged into recombinant lentivirus. The successfully packaged recombinant lentivirus has only one infection ability. CHO-K1 cells are mammalian cells, and the final recombinant protein expressed by CHO-K1 cells is closer to the natural protein structure.

Q4: The lentiviral vector carries the puromycin resistance gene, so the CHO-K1 cells that successfully expressed the recombinant protein have puromycin resistance. After the CHO-K1 cells returned to normal growth, the final concentration of puromycin was 14 µg/mL, and the puromycin was supplemented every 48 h. When all negative controls died, CHO-K1 recombinant cells successfully integrated with exogenous genes TB-Cap, TB-Cap-CD154 and TB-Cap-GM-CSF were obtained (lines 132-134).

Q5: The transferred membranes were blocked by incubating with 5% skim milk in Tris-Buffered Saline Tween-20 (TBST) for 30 min and incubated overnight at 4 C with the anti-PCV2 positve serum primary antibodies at 1:2,000 dilution. HRP-labeled goat anti-porcine antibody (Beyotime Biotechnology, Shanghai, China) was used as the secondary antibody. Protein bands were detected using ECL chemiluminescence system and analyzed by ImageJ Launcher software (lines 140-143).

Q6: We are very sorry that our scientific research is not rigorous enough. The recombinant monoclonal cells were cultured continuously for 25 generations, and the same number of cells (4×105 cells) were taken every 5 generations to test the stability of the fusion protein.

In addition, we revised and strengthened the discussion and English sentences. Once again, thank you very much for your comments and suggestions.

Best regards.

Your sincerely,
Mao Qian

Round 2

Reviewer 1 Report

The current manuscript has been improved. Introduction becomes juicier. Methods are given in more detail and results are clearly presented. However, the abstract could be shortened by briefing the main results and conclusions. Moreover, there are grammar errors here and there in the paper, which should be thoroughly checked and revised before the acceptance of the current manuscript. Some but not all of the errors are listed below.

1, Line 13: please leave a space between "(PCVAD)" and "is".

2, Line 16: please change "stimulate the produce humoral and cellular immunity" to "induce humoral and cellular immunity"

3, Line 17: please change "enhances" to "enhance"

4, Line 26: the grammar of this sentence is not correct. 

5, Line 89: please add "were" in front of the word "preserved".

6, Line 106: "E.coli" should be Italic.

7, Line 208: the volume of primers used was missing.

8, Line 224: the version of the GraphPad Prism software was not mentioned here. 

9, Line 224: "tow-way" should be changed to "two-way"

10, Line 228: "Dentification" should be changed to "Identification"

11, Line 228: "subsection" does not belong to the title, and thus should be removed.

12, Line 251-252: grammar error.

13, Figure 5-9: Please include the interpretation of different asterisk numbers such as (*): p<0.05; (**): p<0.01;(***):p<0.001;(****):p<0.0001

Author Response

Dear Reviewers:

Thank you very much for your comments concerning our manuscript. We tried our best to improve the manuscript and  simplified the abstract and revised the grammar in the new manuscript. These changes will not influence the content and framework of the paper. And here we did not list the changes but marked up in revised paper.

Thank you very much for your hard work and good advice when reviewing my manuscript.

Reviewer 2 Report

Dear Authors,

Thank you for having considered the previous suggestions made and for greatly improving the scientific report.

Congratulations for research outputs and wishes of success.

Author Response

(The authors gave the same response as above.)
